# Point-of-Care and Rapid Tests for the Etiological Diagnosis of Respiratory Tract Infections in Children: A Systematic Review and Meta-Analysis

**DOI:** 10.3390/antibiotics11091192

**Published:** 2022-09-03

**Authors:** Giulia Brigadoi, Andrea Gastaldi, Marco Moi, Elisa Barbieri, Sara Rossin, Annalisa Biffi, Anna Cantarutti, Carlo Giaquinto, Liviana Da Dalt, Daniele Donà

**Affiliations:** 1Department for Women’s and Children’s Health, University of Padua, Via Giustiniani 3, 35128 Padua, Italy; 2Department of Pediatrics, Women’s and Children’s Health, University of Verona, Piazz. Stefani 1, 37126 Verona, Italy; 3Division of Pediatric Infectious Diseases, Department for Women’s and Children’s Health, University of Padua, Via Giustiniani 3, 35128 Padua, Italy; 4Pediatric Emergency Department, Department for Woman and Child Health, University of Padua, Via Giustiani 3, 35128 Padua, Italy; 5Unit of Biostatistics, Epidemiology and Public Health, Department of Statistics and Quantitative Methods, University of Milano-Bicocca, 20126 Milan, Italy; 6National Centre for Healthcare Research and Pharmacoepidemiology, Department of Statistics and Quantitative Methods, University of Milano-Bicocca, 20126 Milan, Italy

**Keywords:** diagnostic stewardship, antimicrobials, children, POCT, film-array, RIDT

## Abstract

Fever is one of the most common causes of medical evaluation of children, and early discrimination between viral and bacterial infection is essential to reduce inappropriate prescriptions. This study aims to systematically review the effects of point-of-care tests (POCTs) and rapid tests for respiratory tract infections on changing antibiotic prescription rate, length of stay, duration of therapy, and healthcare costs. Embase, MEDLINE, and Cochrane Library databases were systematically searched. All randomized control trials and non-randomized observational studies meeting inclusion criteria were evaluated using the NIH assessment tool. A meta-analysis was performed to assess the effects of rapid influenza diagnostic tests and film-array respiratory panel implementation on selected outcomes. From a total of 6440 studies, 57 were eligible for the review. The analysis was stratified by setting and POCT/rapid test type. The most frequent POCTs or rapid tests implemented were the Rapid Influenza Diagnostic Test and film-array and for those types of test a separate meta-analysis assessed a significant reduction in antibiotic prescription and an improvement in oseltamivir prescription. Implementing POCTs and rapid tests to discriminate between viral and bacterial infections for respiratory pathogens is valuable for improving appropriate antimicrobial prescriptions. However, more studies are needed to assess these findings in pediatric settings.

## 1. Introduction

Fever in children is one of the most common causes of medical evaluation in the emergency department (ED) or primary care practices and one of the possible complications in children hospitalized for other reasons [1]. Early recognition of severe infections in acutely ill children is crucial for improving their outcomes. Many physicians prescribe antibiotics, especially broad-spectrum antibiotics, for children with fever while waiting for blood tests and microbiological results to avoid the risk of severe infectious complications. However, it has been demonstrated that up to 50% of antibiotic prescriptions are unnecessary or inappropriate [2], and many children receive broad-spectrum antibiotics for viral infections [3]. This unnecessary use of antibiotics leads to increased antibiotic resistance and healthcare costs. For these reasons, discriminating between viral and bacterial infections is essential. 

In 2007, the Infectious Disease Society of America (IDSA), alarmed by the increasing antimicrobial resistance rate compared to a reduction in newly available antibiotics, introduced the concept of Antimicrobial Stewardship Programs (ASP), defined as a collection of proactive strategies to improve the antibiotic prescription [2]. An integral part of ASPs is represented by diagnostic stewardship, coordinated guidance, and intervention to improve the appropriate use of microbiological diagnostics, which could help discriminate between viral and bacterial infection and guide therapeutic decisions [4]. Indeed, diagnostic tools are required to be accurate and available in a short period of time to modify decisions effectively and have an impact on the choice of treatment, as well as adequately cost saving. 

Rapid tests are designed to give a diagnostic response with a shorter turnaround time than standard analysis (sample culture, serology, etc.). In addition, microbiological point-of-care tests (POCTs) are rapid tests designed to provide a quicker response at the bedside [5]. Indeed, rapid tests and POCTs could reduce inappropriate antibiotic prescriptions and length of stay, improve patient outcomes, and even allow cost savings due to earlier appropriate medical decisions. Many studies have been published on the usability of rapid tests and POCTs for implementing the diagnostic workup in many different settings, especially in adult populations. However, their use is not currently the standard of care for evaluating febrile children.

The primary aim of this review is to summarize the current state of evidence on the impact of rapid tests and POCTs for respiratory tract infections in pediatric settings (ED, inpatient, and outpatient) on changing the antimicrobial prescriptions rate (both antibiotic and oseltamivir), duration of therapy, length of stay, and healthcare costs in high and low–middle income countries.

## 2. Results

A total of 6440 studies were retrieved from the database search, excluding duplicates. After title and abstract screening, 114 were eligible for full-text reading and 52 were included in this review. In addition, five studies were added from the manual reviewing of reference lists for a total of 57 papers. The selection process is summarized in Figure 1 [6].

The title, authors, publication year, study design, country, study period, setting, number of patients included, type of rapid test or POCT, and types of outcomes considered are summarized in Table 1. The results of each study for the different outcomes are reported in Table 2.

Most of the studies were published after 2007 (47/57, 82.5%), with 18 studies (10/57, 17.5%) between 2020 and 2021. Fifty-three studies (53/57, 93.0%) were conducted in high-income countries [7], and almost 80% of these articles described the implementation of a rapid test or POCT in North America (25/57, 43.9%) or Europe (19/57, 33.3%). Only eight studies (8/57, 14.0%) were conducted in Asia. The geographical distribution of articles is shown in Figure 2.

Thirteen studies were multicenter: more than half were set in Europe (8/13, 61.5%) and the other 5 in North America (5/13, 38.5%).

The distribution of studies according to their design was: 8 randomized controlled trials (14.0%), 27 observational studies, including retrospective or prospective observational studies and control–case studies (47.4%), 22 quasi-experimental studies including before and after studies and interrupted time series studies (38.6%)

The most frequent rapid tests or POCTs implemented were the Rapid Influenza Diagnostic Test (RIDT) (22/57, 38.6%) and Film Array–Respiratory Panel (FA-RP) (22/57, 38.5%), followed by Strep A rapid tests (7/57, 12.3%) and others testing Respiratory Syncytial Virus (RSV) or the combination of RSV and influenza. Only one study focused on the implementation of a rapid PCR test for *Mycoplasma pneumoniae*.

**Table 1 antibiotics-11-01192-t001:** Characteristics of studies included in the review.

Authors	Year of Publication	Study Design	Study Location	Study Period	Single or Multi-Center	Age	Care Setting	N° Patients	Type of Test	Outcomes					NIH Tool
										PR	DOT	LOS	COST	PO	
Nitsch-Osuch et al. [8]	2017	QE	POL	January 2015–March 2015 January 2016–March 2016	Single	<5 years	Inpatient	115	RIDT	X				X	FAIR
Vecino-Ortiz et al. [9]	2018	QE	GBR	2013–2014 2014–2015	Single	Children	Inpatient	574	RIDT, Rapid VRS test	X		X	X	X	FAIR
Abanses et al. [10]	2006	RCT	USA	December 2002–March 2003	Single	3–36 months	ED	1007	RIDT	X		X	X		FAIR
Diallo et al. [11]	2019	OR	FRA	2013–2015	Single	1 month–18 years	ED	241	RIDT			X			FAIR
Noyola et al. [12]	2000	OR	USA	July 1995– June 1997	Single	Children	ED	1530	RIDT	X	X	X		X	GOOD
Özkaya et al. [13]	2009	OP	TUR	November 2006–March 2007	Single	3–14 years	ED	97	RIDT	X					FAIR
Bonner et al. [14]	2003	RCT	USA	January 2002–March 2002	Single	2 months–21 years	ED	391	RIDT	X		X		X	GOOD
Cantais et al. [15]	2019	OP	FRA	January 2016–March 2016	Single	0–16 years	ED	514	RIDT	X					FAIR
Iyer et al. [16]	2006	Quasi-RCT	USA	January 2003–March 2003 December 2003–January 2004	Single	2–24 months	ED	700	RIDT	X		X	X		FAIR
Jacob et al. [17]	2020	OR	AUS	August 2017–September 2017	Single	<16 years	ED	1451	RIDT	X		X			FAIR
Jun et al. [18]	2016	QE	KOR	December 2008–January 2009 February 2013–March 2013	Single	<16 years	ED	342	RIDT	X		X			FAIR
Li-Kim-Moy et al. [19]	2016	OR	AUS	2009	Single	<18 years	ED	364	RIDT	X		X		X	FAIR
Sharma et al. [20]	2002	OR	USA	November 1998–March 1999 November 1999–March 2000	Single	2 months–2 years	ED	72	RIDT	X		X			FAIR
Patel et al. [21]	2020	QE	USA	November–March of 2014 to 2017 November–March of 2017 to 2018	Single	3 months–18 years	ED	5307	RIDT	X		X		X	FAIR
Pierron et al. [22]	2008	OP	FRA	January 2007–March 2007	Single	1 month–6 years	ED	177	RIDT	X					FAIR
Benito-Fernandez et al. [23]	2006	OP	ESP	November–December 2003 December 2004–February 2005	Single	0–14 years	ED	206	RIDT	X		X			FAIR
Poehling et al. [24]	2006	RCT	USA	2002–2004	Multi-center	<5 years	ED + outpatient	468	RIDT	X					GOOD
Jennings et al. [25]	2009	OP	DEU	January 2007– April 2007	Multi-center	1–12 years	Outpatient	16907	RIDT	X				X	FAIR
Nitsch-Osuch et al. [26]	2013	OP	POL	2009/2010–2010/2011	Multi-center	<5 years	Outpatient	256	RIDT	X				X	FAIR
Van Esso et al. [27]	2019	OP	ESP	2016–2017	Multi-center	0–6 years	Outpatient	1170	RIDT	X					POOR
Cohen et al. [28]	2007	OP	FRA	2004–2005	Multi-center	Children	Outpatient	602	RIDT	X				X	FAIR
de La Rocque et al. [29]	2009	OP	FRA	December 2006–April 2007	Multi-center	Children	Outpatient	695	RIDT	X				X	GOOD
Keske et al. [30]	2018	QE	TUR	January 2015–December 2016	Single	<16 years	Inpatient	258	FA-RP	X	X				FAIR
Kitano et al. [31]	2019	QE	JPN	March 2018 – April 2019 March 2012–March 2018	Single	*nd*	Inpatient	1281	FA-RP		X	X	X		FAIR
Lee et al. [32]	2020	QE	USA	December 2009–July 2012 August 2012 – June 2016	Single	<18 years	Inpatient	5142	FA-RP	X		X		X	FAIR
Reischl et al. [33]	2020	QE	DEU	February 2016–February 2017 February 2017–April 2018	Single	0–2 years	Inpatient	786	FA-RP	X	X	X		X	FAIR
Schulert et al. [34]	2013	OR	USA	August 2009–December 2010	Single	0–14 years	Inpatient	790	FA-RP		X	X			FAIR
Subramony et al. [35]	2016	QE	USA	June 2010 – June 2012 October 2012–May 2014	Single	0–18 years	Inpatient	4779	FA-RP		X				FAIR
Walls et al. [36]	2016	OR	NZL	Winter months 2012–2015	Single	3 months–5 years	Inpatient	237	FA-RP	X	X				FAIR
Yoshida et al. [37]	2021	QE	JPN	December 2017–November 2018 March 2019–February 2020	Single	0–18 years	Inpatient	181	FA-RP	X	X	X			FAIR
McCulloh et al. [38]	2013	OR	USA	October 2009–April 2010 October 2010–April 2011	Single	0–18 years	Inpatient	1727	FA-RP	X	X			X	FAIR
McFall et al. [39]	2017	OR	USA	November 2012–August 2015	Single	1–3 months	Inpatient	176	FA-RP		X	X			FAIR
Iroh Tam et al. [40]	2017	OR	USA	January 2011 – April 2015	Multi-center	<18 years	ED + inpatient	1625	FA-RP	X		X	X		FAIR
Kim et al. [41]	2021	QE	KOR	November 2015–June 2016 November 2016–July 2018	Single	1 month–18 years	ED + inpatient	915	FA-RP	X	X	X			FAIR
Rogers et al. [42]	2015	QE	USA	November 2011–January 2012 November 2012–January 2013	Single	3 months–21 years	ED + inpatient	1136	FA-RP	X	X	X			FAIR
Busson et al. [43]	2019	QE	BEL	February 2016–March 2016	Single	Adults and children, separated data	ED	142	FA-RP	X		X		X	FAIR
Byington et al. [44]	2002	OR	USA	December 2000–February 2001 December 2001–January 2002	Single	<19 years	ED	338	FA-RP	X	X				FAIR
Crook et al. [45]	2020	QE	USA	January 2011–December 2014 January 2015– April 2018 May 2018– June 2019	Single	<90 days	ED	5317	FA-RP	X	X	X			FAIR
Dimopoulou et al. [46]	2020	OP	GRC	February 2019	Single	0–16 years	ED	80	FA-RP	X				X	FAIR
Rao et al. [47]	2021	RCT	USA	December 2018–November 2019	Single	1 month–18	ED	920	FA-RP	X		X		X	GOOD
Echavarría et al. [48]	2018	RT	ARG	April–November 2016 April–October 2017	Single	2 months–6 years	ED	156	FA-RP	X		X		X	GOOD
May et al. [49]	2019	RCT	USA	December 2016–April 2018	Single	1–17 years	ED	71	FA-RP	X		X		X	GOOD
Wishaupt et al. [50]	2011	RCT	NLD	November 2007–May 2008 October 2008–March 2009	Single	0–12 years	ED + outpatient	583	FA-RP		X	X			GOOD
Beal et al. [51]	2020	QE	USA	January 2018–January 2019	Multi-center	<21 years	Outpatient	430	FA-RP	X		X		X	FAIR
Thibeault et al. [52]	2007	OR	CAN	Winter 2001–2002 and 2002–2003	Multi-center	0–3 years	Inpatient	448	Rapid RSV test	X	X				FAIR
Schnell et al. [53]	2017	OR	USA	November–March of 2008–2013	Single	2 months–2 years	ED	713	Rapid RSV test	X		X			FAIR
O’ Callaghan et al. [54]	2019	QE	AUS	May 2017– August 2017 May 2018–August 2018	Single	Children	ED	642	Rapid PCR for influenza and RSV	X		X			FAIR
Mitchell et al. [55]	2018	QE	USA	November 2014–March 2015	Single	<18 years	ED	2171	Rapid PCR for influenza and RSV					X	FAIR
Schneider et al. [56]	2018	OP	DNK	February 2018–July 2018	Multi-center	0–18 years	ED	180	Rapid PCR for influenza and RSV	X		X			FAIR
Hayashi et al. [57]	2018	OP	JPN	May 2016– April 2017	Single	<18	Inpatient	375	Mycoplasma PCR	X					FAIR
Ayanruoh et al. [58]	2009	QE	USA	September 2005–September 2007	Single	3–18 years	ED	8280	RSTs	X					FAIR
Bird et al. [59]	2021	QE	GBR	October–November 2014 August–November 2015 September–November 2016	Single	6 months–16 years	ED	605	RSTs	X					FAIR
Halverson et al. [60]	2011	QE	USA	October 2006–December 2009	Multi-center	nd	ED	nd	RSTs			X			POOR
Kose et al. [61]	2016	QE	TUR	February 2012–May 2014	Single	3–14 years	ED	223	RSTs	X			X		FAIR
Małecki et al. [62]	2017	QE	POL	October 2013– April 2014	Multi-center	2–15 years	Outpatient	1307	RSTs	X	X	X	X		FAIR
Maltezou et al. [63]	2008	OP	GRC	December 2005–June 2006 September 2006–June 2007	Multi-center	2–14 years	Outpatient	820	RSTs	X	X	X	X		FAIR
Rao et al. [64]	2019	OP	USA	Fall-winter 2016–2017	Single	3–18 years	Outpatient	275	RSTs and PCR Strep A test	X					FAIR

PR = prescription rate; DOT = days of therapy; LOS = length of stay; PO = prescription of oseltamivir; NIH = National Institutes of Health; OP = observational prospective; RCT= randomized control trial; RT= randomized trial; OR = observational retrospective; QE = quasi-experimental; ED = emergency department; RIDT = Rapid Influenza Diagnostic Test; FA-RP = Film Array–Respiratory Panel; RSV = Respiratory Syncytial Virus; RSTs = Rapid Streptococcal Tests; PCR = polymerase chain reaction; regarding outcomes, X indicates which outcome is considered in each study.

**Table 2 antibiotics-11-01192-t002:** Outcomes of the studies included in the review.

Authors and Year of Publication	N° Patients	Type of Test	Outcomes					NIH Tool
			PR	DOT	LOS	COST	PO	
Nitsch-Osuch et al., 2017 [8]	115	RIDT	Antibiotic therapy was statistically more frequently administered when RIDT was not available (93% vs. 64%; *p* < 0.05)				Oseltamivir was statistically more prescribed when RIDT was available (64% of patients with influenza received an antiviral, none of the children received an antiviral without RIDT)	FAIR
Vecino-Ortiz et al., 2018 [9]	574	RIDT, rapid VRS test	No significant differences in the antibiotic prescription rate between periods in those positive for influenza and negative for both influenza and RSV		There was no significant difference between the periods for the total length of stay (median = 2 days for both periods, *p* = 0.23)	Reductions in the average reimbursement charge for patients with a negative influenza and RSV test. No change in reimbursement for patients with proven influenza or RSV infection	Small but significant increase in the cost of drugs between periods 1 and 2 for admissions in which the patients were positive for influenza and/or RSV	FAIR
Abanses et al., 2006 [10]	1007	RIDT	Significant reduction in antibiotic prescription in those testing positive for influenza (215 vs. 102, RR 0.85, CI 95% 0.7–1.02)		Time in ED was significantly less in the intervention group (195 vs. 156 min; 95% CI for the difference, 19–60)	Total medical charges were significantly less in the intervention group (USD 666 vs. USD 393; 95% CI for the difference, 153–392)		FAIR
Diallo et al., 2019 [11]	241	RIDT			The mean length of stay in the PED was significantly lower in the positive RDT group: 4.0 h vs. 7.4 h (*p* < 10^−6^)			FAIR
Noyola et al., 2000 [12]	1530	RIDT	Patients discharged from the ED with a positive influenza test were less likely to receive antibiotics than those with a negative test (20% vs. 53%; *p* < 0.04) Patients admitted to the hospital with a positive EIA test were as likely to receive antibiotics as those without a rapid diagnosis	Duration of antibiotic administration was significantly shorter in the group with a positive influenza test (3.5 vs. 5.4 days; *p* = 0.03)	Patients with a positive influenza test were more likely to have a shorter duration of admission than the control group (4.3 mean days versus 7.4; *p* = 0.02)		Patients with a positive influenza test were more likely to receive antiviral therapy than the control group (25% vs. 0 and 1.8%; *p* < 0.001)	GOOD
Özkaya et al., 2009 [13]	97	RIDT	Patients in group testing prior to prescription were less likely to be prescribed antibiotics when compared to those in the group in which rapid testing was not considered for prescription (32% vs. 100%, respectively, *p* < 0.0001)					FAIR
Bonner et al., 2003 [14]	391	RIDT	Reduction in antibiotic prescription in the group in which the results of the rapid test was known in comparison to the group of unknown (7/96 vs. 26/106, *p* < 0.001); no difference between negative test and negative unaware test (27/97 vs. 27/92, *p* = 0.0818)		Reduction in the length of stay in the group in which the results of the rapid test were known in comparison to the group of unknown (25 min vs. 49 min, *p* < 0.001); no difference between the negative test and negative unaware test (45 min vs. 42 min, *p* = 0.549)		Increase in antiviral prescription in the group in which the results of the rapid test were known in comparison to the group of unknown (18/96 vs. 7/106, *p* = 0.02); no difference between negative test and negative unaware test (0/97 vs. 2/92, *p* = 0.236)	GOOD
Cantais et al., 2019 [15]	514	RIDT	Reduction in antibiotic prescriptions, 60 of 245 patients (24.5%) received antibiotics in the DIA negative group, versus 25 of 262 (9.5%) in the positive one (*p* < 0.001)					FAIR
Iyer et al., 2006 [16]	700	RIDT	No significant differences were demonstrated between the POCT and standard test groups with respect to antibiotic prescription		No significant differences were demonstrated between the POCT and standard test groups with respect to lengths of stay	No significant differences were demonstrated between the POCT and standard test groups with respect to visit-associated costs		FAIR
Jacob et al., 2020 [17]	1451	RIDT	Antibiotics were used more in patients with ILI with no RIDT (15.2% in the ILI group vs. 2.7% in the laboratory RIDT group and 11.2% in the ED RIDT group; *p* < 0.0001)		Patients for whom RIDT was performed at the laboratory had a shorter length of stay when compared to patients for whom RIDT was performed bedside in the ED (4.7 and 5.3 h, respectively; *p* < 0.0001),			FAIR
Jun et al., 2016 [18]	342	RIDT	Reduction in antibiotic prescription in RAT-positive patients, after the 2009 influenza pandemic (none of the pediatric patients received antibiotics)		The duration of ER stay in discharged patients was 268.9 ± 144.2 min in patients with the use of a RAT kit and 210.5 ± 205.3 min in patients with no use of a RAT kit after the 2009 influenza pandemic			FAIR
Li-Kim-Moy et al., 2016 [19]	364	RIDT	Compared with standard testing (n = 65), children diagnosed by positive POCT (n = 236) had a reduction in antibiotic use (odds ratio 0.42, *p* = 0.003)		Compared with standard testing, children diagnosed by positive POCT had a shorter median hospital LOS by 1 day (*p* = 0.006). POCT did not decrease LOS in ED		Compared with standard testing, children diagnosed by positive POCT had increased antiviral prescription (odds ratio 3.1, *p* < 0.001)	FAIR
Sharma et al., 2002 [20]	72	RIDT	Fewer patients in the early diagnosis group received ceftriaxone sodium compared with the late diagnosis group (2% vs. 24%, *p* = 0.006)		No significant differences were demonstrated between the POCT and standard test groups with respect to lengths of stay			FAIR
Patel et al., 2020 [21]	5307	RIDT	There was no significant difference in rates of antibiotics used		The median LOS decreased from 239 min in the pre-POC period to 232 min in the post-POC period (*p* < 0.05)		There were increased rates of oseltamivir used in the post-POC period (21.2% vs. 13.3%, *p* < 0.05	FAIR
Pierron et al., 2008 [22]	177	RIDT	There was not any significant difference concerning antibiotic prescriptions					FAIR
Benito-Fernandez et al., 2006 [23]	206	RIDT	There was a significant reduction in the use of antibiotics (38.5% vs. 0%, *p* < 0.01)		There was a significant reduction in the mean length of stay in the ED (192.9 versus 116.2 min) (*p* < 0.01)			FAIR
Poehling et al., 2006 [24]	468	RIDT	There was no difference in antibiotic prescribing					GOOD
Jennings et al., 2009 [25]	16907	RIDT	Antibiotics were less commonly prescribed for children who were influenza positive by rapid test (3.5% (271/7685) versus 17.2% (125/725) for symptom assessment alone)				The antiviral oseltamivir was prescribed for 24.6% (178/725) of children who were influenza positive by symptom assessment alone and 60.1% (4618/7685) of children who were influenza positive by rapid test	FAIR
Nitsch-Osuch et al., 2013 [26]	256	RIDT	Antibiotics were administered more often in the control group compared with the rapid test group (respectively, for 16% vs. 7%). No child with a positive result of RIDT was prescribed an antibiotic				The antiviral treatment (oseltamivir) was prescribed only for four children with positive results of RIDT	FAIR
Van Esso et al., 2019 [27]	1170	RIDT	Influenza-confirmed patients received fewer antibiotics during the 10 days after influenza diagnosis but not statistically significant compared with the groups with a clinical diagnosis of influenza without a microbiologic confirmation					POOR
Cohen et al., 2007 [28]	602	RIDT	The antibiotic prescription was overall low (9.5% with RIDT vs. 3.9% without RIDT, *p* = 0,008), and primarily when the result of RIDT was negative (15.7% if RIDT– vs. 4.3% if RIDT+, *p* = 0.0003)				The pediatricians using RIDT prescribed with positive tests more oseltamivir (68.5 vs. 1.9%, *p* < 0.0001)	FAIR
de La Rocque et al., 2009 [29]	695	RIDT	The RIDT+ group received antibiotics in 7.6% of cases, RIDT− in 18.5% (*p* < 0.0001)				The RIDT+ group received an antiviral in 64.7% and the RIDT− group received no antiviral (*p* < 0.0001)	GOOD
Keske et al., 2018 [30]	258	FA-RP	Significant decrease in antibiotic use (44.5% in 2015 and 28.8% in 2016, *p* = 0.009)	The duration of antibiotic use after the detection of virus was significantly decreased in children (*p* < 0.001)				FAIR
Kitano et al., 2019 [31]	1281	FA-RP		The DOT/case was 12.82 vs. 8.56 (*p* < 0.001), in the rapid antigen test and mPCR groups, respectively	The LOS was 8.18 vs. 6.83 days (*p* = 0.032) in the rapid antigen test and mPCR groups, respectively	The total costs during admissions were 258,824 (USD 2331.7) and 243,841 (USD 2196.8)/case, in the rapid antigen test and mPCR groups, respectively		FAIR
Lee et al., 2020 [32]	5142	FA-RP	Patients tested with RP were less likely to receive empiric antibiotics (OR: 0.45; *p* < 0.001; 95% CI: 0.39, 0.52) compared to RVP patients	Patients tested with RP had a shorter duration of empiric broad-spectrum antibiotics (6.4 h vs. 32.9 h; *p* < 0.001) compared to RVP patients			RP influenza patients had increased oseltamivir use post-test compared to RVP influenza patients (OR: 13.56; *p* < 0.001; 95% CI: 7.29, 25.20).	FAIR
Reischl et al., 2020 [33]	786	FA-RP	The binary logistic regression analysis shows no significant (*p* = 0.784) impact of the FA-RP or the multiplex RT-PCR on the antibiotic treatment	The diagnostic method, FA-RP (8.6 days) or multiplex RT-PCR (9.1 days), showed no significant (*p* = 0.592) impact on the duration of antibiotic treatment in the linear logistic regression analysis	The mean hospital length of stay for both study groups was 4.7 days. The diagnostic method, FA or multiplex RT-PCR, showed no significant impact on the length of hospital stay in the linear regression analysis.		No significant difference in antiviral prescriptions	FAIR
Schulert et al., 2013 [34]	790	FA-RP		The median duration of IV antibiotics for patients with a positive RVP was 55 h, compared with 96 h for patients with a negative RVP (*p* = 0.03)	The median length of stay for patients with a positive RVP was 3 days, compared with 4 days for patients with a negative RVP (*p* = 0.057)			FAIR
Subramony et al., 2016 [35]	4779	FA-RP		Subjects in the mPCR group received fewer days of antibiotics than subjects in the non-mPCR group (4 vs. 5 median antibiotic days, *p* < 0.01)				FAIR
Walls et al., 2016 [36]	237	FA-RP	A significantly larger proportion of children who had an NPS sample taken (42/146, 36%) received no empiric antibiotics compared to children who did not have a sample taken (7/91, 7.7%, *p* < 0.001)	Of those who did have an NPS sample taken, 17 of 146 (11.6%) had their antibiotics discontinued prior to or at the time of discharge compared with only 3 of 91 (3.3%) of those who did not have an NPS sample (*p* < 0.025)				FAIR
Yoshida et al., 2021 [37]	181	FA-RP	We did not observe differences in the use of antibiotics between the pre- and post-mPCR periods (*p* = 0.14)	We did not observe differences in the duration of antibiotic usage between the pre- and post-mPCR periods (*p* = 0.45)	We did not observe differences in the length of stay between the pre- and post-mPCR periods (*p* = 0.94)			FAIR
McCulloh et al., 2013 [38]	1727	FA-RP	Children with a positive RVP test result received antibiotics less often (363 of 703 (51.6%) vs. 71 of 106 (67.0%); *p* = 0.003)	In total, 21 of 348 (6.0%) children who were positive for a viral pathogen by RVP had antibiotics discontinued within 24 h after RVP test results were available, but no children with negative RVP results had antibiotics subsequently stopped			Children with a positive RVP test result received oseltamivir more often (76.9% vs. 18%; *p* < 0.001)	FAIR
McFall et al., 2017 [39]	176	FA-RP		Duration of antimicrobial consumption was significantly decreased in patients with a positive FA-RP compared to infants with a negative test (mean rank 2.8 vs. mean rank 5.2 days), *p* < 0.001)	For all infants with a positive FA-RP result, LOS was significantly decreased compared with infants with a negative FA-RP result (5.7 vs. 10.4 days, *p* = 0.017)			FAIR
Iroh Tam et al., 2017 [40]	1625	FA-RP	No difference in antibiotic prescription for all types of antibiotics		Patients with a positive test from RVPP had shorter LOS (*p* = 0.0503)	Hospital charges for patients with a positive test from RVPP were lower, but not significantly so	No difference in antiviral prescription (*p* = 0.76)	FAIR
Kim et al., 2021 [41]	915	FA-RP	FA-RP reduced intravenous (IV) antibiotic use (*p* = 0.002)	FA-RP reduced the duration of intravenous (IV) antibiotic use, for pediatric patients (*p* < 0.001)	FA-RP reduced the lead time, waiting time, turnaround time, and length of hospital stay (*p* = 0.004)			FAIR
Rogers et al., 2015 [42]	1136	FA-RP	The number of patients receiving antibiotics and the inpatient LOS did not differ in the 2 groups	Duration of antibiotic use decreased for patients in the post-FA-RP group by 0.4 days (*p* = 0.003)	The LOS in the ED increased by 26 min in the post-RRP group (*p* = 0.002)			FAIR
Busson et al., 2019 [43]	142	FA-RP	Results from the FilmArray Respiratory Panel do not appear to impact antibiotic prescription		The mean length of stay was not significantly different between the two groups (3.9 days for the group with a positive FA result vs. 5.2 days for the group with a negative FA result; *p* = 0.286)		No difference in oseltamivir prescription	FAIR
Byington et al., 2002 [44]	338	FA-RP	Test-positive patients had fewer discharge prescriptions for oral antibiotics (37% vs. 52%, *p* = 0.02) when compared with test-negative patients. Intravenous antibiotics were initiated less often for test-positive patients during the second winter season than during the first (26% vs. 44%, *p* = 0.008)	Test-positive patients had fewer days using intravenous antibiotics (2.4 vs. 4, *p* = 0.04), fewer days using oral antibiotics (0.25 vs. 2.5, *p* = 0.04), when compared with test-negative patients				FAIR
Crook et al., 2020 [45]	5317	FA-RP	Following introduction of mPCR testing, the percentage of patients who did not receive antimicrobials increased from 32.4% to 43.1% (difference, 10.8%; 95% CI, 6.5–15%)	Median antibiotic duration decreased by 0.47 days (95% CI, 0.16–0.51)	There was a significant reduction in LOS (*p* < 0.001)			FAIR
Dimopoulou et al., 2020 [46]	80	FA-RP	The implementation of a rapid molecular test had no impact on antibacterial prescription (10% vs. 13.3%).				The implementation of a rapid molecular test had no impact on antiviral prescription	FAIR
Rao et al., 2021 [47]	920	FA-RP	In the intention-to-treat intervention group (result known), children were more likely to receive antibiotics (relative risk (RR), 1.3; 95% CI, 1.0–1.7) compared to the control group (result not known)		No significant differences in length of stay between the two groups		No significant differences in antiviral prescribing between the two groups	GOOD
Echavarría et al., 2018 [48]	156	FA-RP	Diagnosis with FA-RP was associated with significant changes in medical management including withholding antibiotic prescriptions (OR:12.23, 95%CI:1.56–96.09)		The median LOS was lower for the FA-RP group (4 days) than the control group (10 days) although the difference was not statistically significant (*p* = 0.382)		Oseltamivir usage was very low and no significant changes in treatment with the drug were observed between the two study groups	GOOD
May et al., 2019 [49]	71	FA-RP	In total, 20 (22%) RP patients and 33 (34%) usual-care patients received antibiotics during the ED visit (–12%; 95% confidence interval, –25% to 0.4%; *p* = 0.06/0.08)		There was no significant difference in length of ED stay, or hospital stay among admitted patients between the 2 groups		No significant difference in antiviral prescription (+3% (–5% to –10%) 0.53/0.61)	GOOD
Wishaupt et al., 2011 [50]	583	FA-RP		Mean durations of antibiotic treatment, if antibiotic treatment was started, did not differ significantly between the groups	There was a trend toward a shorter length of hospital stay in the intervention group, but the difference was not statistically significant			GOOD
Beal et al., 2020 [51]	430	FA-RP	Appropriate treatment occurred for 93.6% of patients when the FA-RP was performed (Clinic A) versus 87.9% of patients who had only antigen tests performed (Clinic B, *p* = 0.0445)		Utilization of FA-RP testing also significantly reduced appointment duration time (48.0 versus 54.9 min, *p* = 0.0009)		Patients tested with FA-RP received less oseltamivir compared to children tested with an antigen test (*p* = 0.0018)	FAIR
Thibeault et al., 2007 [52]	448	Rapid RSV test	There was no significant difference between children with positive and negative RSV RADT results in the percentage receiving IV antibiotics only (10% versus 7%, *p* = 0.61); PO antibiotics only (38% versus 28%, *p* = 0.17); or both PO and IV antibiotics (52% versus 65%, *p* = 0.12)	At 24, 48, or 72 h, stopping or switching of IV antibiotics was not influenced by the RADT result in any of the four strata combining age and presence of pneumonia (<3 months and no pneumonia; <3 months without pneumonia; ≥3 months with pneumonia; ≥3 months with- out pneumonia)				FAIR
Schnell et al., 2017 [53]	713	Rapid RSV test	Antibiotic administration within the ED did not differ between those testing positive for RSV versus those testing negative		The mean time in the department was not statistically significant between the 2 groups at 174.1 (SD, 89.8) minutes for the RSV-negative group and 165.2 (SD, 84.6) minutes for the RSV-positive group			FAIR
O’ Callaghan et al., 2019 [54]	642	Rapid PCR for influenza and RSV	26.3% of positive influenza A/B RSV patients were treated with antibiotics in 2017, whereas 21.3% were treated with antibiotics in 2018 (*p* = 0.45)		According to time to discharge in the ED, there were no differences between positive and negative patients (*p* = 0.85)			FAIR
Mitchell et al., 2018 [55]	2171	Rapid PCR for influenza and RSV					Analysis of the post-implementation period revealed a significantly lower percentage (14.3%, *p* < 0.001) of negative patients receiving antiviral therapy compared to the pre-implementation period, with no difference in prescription of oseltamivir in those testing positive	FAIR
Schneider et al., 2018 [56]	180	Rapid PCR for influenza and RSV	A positive POCT result significantly reduced antibiotic prescription (*p* < 0.0011)		A positive POCT result significantly reduced median hospitalization time by 14.2 h for children			FAIR
Hayashi et al., 2018 [57]	375	Mycoplasma PCR	Antimicrobial agents for atypical pathogens (macrolides, tetracyclines, or quinolones) were prescribed in 97.3% (217/223) at the initial evaluation, and their prescription rates increased to 99.1% (221/223) during management					FAIR
Ayanruoh et al., 2009 [58]	8280	RSTs	Rapid strep testing was associated with a lower antibiotic prescription rate for children with pharyngitis (41.38% for those treated in the pre-RST phase versus 22.45% for those treated in the post-RST phase; *p* < 0.001)					FAIR
Bird et al., 2021 [59]	605	RSTs	The baseline prescribing rate was 79%, whereas rates after intervention were 24% in 2015 and 28% in 2016					FAIR
Halverson et al., 2011 [60]	nd	RSTs			Implementation of POCT was shown to provide a statistically significant drop in LOS of patients who had group A strep testing performed on them, discharging them 25–30 min faster than other patients on average			POOR
Kose et al., 2016 [61]	223	RSTs	Antibiotic prescription decreased by 42.6% after learning RST results			Antibiotic costs in non-Group A streptococcus pharyngitis, Group A streptococcus pharyngitis, and all subjects’ groups decreased by 80.8%, 48%, and 76.4%, respectively		FAIR
Małecki et al., 2017 [62]	1307	RSTs	Reduction in antibiotic use by 5.1%.			The anticipated cost of treatment decreased by 17%		FAIR
Maltezou et al., 2008 [63]	820	RSTs	Pediatricians without access to laboratory tests were more likely to prescribe antibiotics compared with pediatricians with access to tests (72.2% versus 28.2%, *p* < 0.001)					FAIR
Rao et al., 2019 [64]	275	RSTs and PCR Strep A test	The use of POC PCR resulted in the appropriate use of antibiotics in 97.1% of cases compared with 87.5% of cases for the standard of care, RST plus confirmatory bacterial culture (*p* = 0.0065)					FAIR

PR = prescription rate; DOT = days of therapy; LOS = length of stay; PO = prescription of oseltamivir; NIH = National Institutes of Health; RIDT = Rapid Influenza Diagnostic Test; FA-RP = Film Array–Respiratory Panel; RSV = Respiratory Syncytial Virus; RSTs = Rapid Streptococcal Tests; PCR = polymerase chain reaction; regarding outcomes, X indicates which outcome is considered in each study; RP = respiratory panel; RVP = respiratory viral panel; NPS = nasopharyngeal swab; RVPP = respiratory virus PCR panel.

### 2.1. Implementation Setting

More than three-quarters of the studies described the implementation of rapid tests or POCTs in a hospital setting (46/57, 80.7%). More than half were exclusively conducted in the ED (29/46, 63.0%), and 14 studies (14/46, 30.4%) referred only to hospital wards. Three studies (3/46, 6.52%) described the implementation of rapid tests or POCTs both in ED and hospital wards. Nine studies (9/57, 15.8%) were conducted in the outpatient setting and two studies both in the outpatient and ED settings.

The implementation of different rapid tests and POCTs according to different settings is summarized in Table 1.

#### 2.1.1. Hospital Setting: Inpatient—Emergency Department

Most of the studies on the implementation of FA-RP were conducted in a hospital setting (20/22, 90.9%), 7 studies in the ED (7/20, 35.0%) and 10 studies in different hospital wards (10/20, 50.0%); 3 studies were conducted both in the ED and hospital wards (3/20, 15.0%). More than half of the studies on the implementation of RIDT were conducted in a hospital setting (16/22, 72.7%), almost all in the ED (14/16, 87.5%). Rapid tests for RSV, Mycoplasma, or rapid tests combined for RSV and influenza were implemented only in the hospital setting (three studies about the RSV rapid test, three studies for the combined test, one study about Mycoplasma). Almost all the studies on the implementation of rapid tests or POCTs in hospital wards were conducted in high-income countries.

#### 2.1.2. Outpatient—Primary Care

The most implemented POCT in the outpatient setting was RIDT (5/9, 55.6%), followed by Strep A test (3/9, 33.3%). Only one study described the implementation of FA-RP in the outpatient setting.

Almost all the studies about POCTs in outpatient settings were conducted in Europe (7/9, 77.8%), and only two in the USA (2/9, 22.2%).

### 2.2. Implementation of Rapid Tests or POCTs and Outcomes

In Figure 3, the distribution of studies for each outcome is reported, stratified for setting and country.

Considering the studies’ variety, separate meta-analyses for primary and secondary outcomes were conducted for the two types of rapid tests/POCTs with the most consistent data: film-array respiratory panel FA-RP and RIDT, including a total of 15 articles and 21 articles, respectively.

#### 2.2.1. Antibiotics Prescription

More than three-quarters of the studies reported as their main outcome the change in antibiotics prescription after rapid tests or POCT implementation (49/57, 86.0%), with almost two-thirds (32/49, 65.3%) assessing a statistically significant reduction.

All the articles that focused on the change in antibiotic prescription after the introduction of the Strep A test reported a significant reduction in the antibiotic use, both in inpatient and outpatient settings.

On the contrary, the two articles about the implementation of a rapid test for RSV showed no difference in the antibiotic prescriptions between patients testing positive and those testing negative, both in the ED and hospital wards. One of the studies on the rapid test for RSV combined with influenza reported a statistically significant reduction in antibiotic use in children with a positive test, while another showed a slight reduction in antibiotic prescription. The study about the implementation of a rapid PCR for Mycoplasma pneumoniae showed increased appropriateness in prescribing antibiotics for atypical pathogens in those patients testing positive.

Separate meta-analyses for FA-RP and RIDT are reported in Figure 4 and Figure 5, and subgroup analyses are reported in Appendix A. It was not possible to conduct the pre-planned metanalysis by age groups due to the lack of data.

When comparing the use of FA-RP vs. standard test, an overall reduction in antibiotic prescriptions was noted (OR = 0.59, 95% CI 0.37–0.92, *p* = 0.02) [32,33,37,40,41,46,51], however, with a high heterogeneity (I^2^) of 89% (Figure 4A). In a subgroup analysis stratified by countries, this reduction was more significant in the studies conducted in the US compared to Europe and East Asia (see Appendix A). However, when comparing FA-RP vs. clinical diagnosis [30,36,38,42,47,49], prescription rate did not significantly decrease (OR = 0.75, 95% CI 0.49–1.17, *p* = 0.21; I^2^ = 86%) (Figure 4B). A subgroup analysis stratified for different types of study is shown in Appendix A.

No significant reduction in antibiotic prescription (OR = 0.64, 95% CI 0.36–1.15, *p* = 0.14; I^2^ = 95%) was noted when comparing the use of RIDT vs. clinical diagnosis (Figure 5A) [9,10,13,17,18,21,24,25,26,28]. One of the studies (Ozkaya et al.) seems to have a more evident reduction in antibiotic prescribing compared to the other studies. This study, conducted in Turkey, involved only 97 children and all the patients not tested with the rapid antigen detection test received antibiotic therapy. We evaluated this study as “fair” and it had a low weight in our metanalysis.

However, when stratified by geographical location (continent, nation, or single state based on the number of articles included in the analysis), a trend toward fewer prescriptions emerged in Europe and East Asia with respect to the US (see Appendix A).

Comparing a positive result of RIDT vs. a negative result, a significant reduction in the prescriptions rate was noted (OR = 0.35, 95% CI 0.21–0.57, *p* < 0.0001; I^2^ = 61%) (Figure 5B) [12,15,22,23,28,29]. A positive RIDT was also associated with a significant reduction in antibiotics prescriptions with respect to a positive standard test [8,12,19,20,27] (OR = 0.43, 95% CI 0.25–0.74, *p* = 0.003; I^2^ = 31%) (Figure 5C). This evidence seemed stronger in Europe than in the US (see Appendix A). A subgroup analysis stratified for different types of study is shown in Appendix A.

#### 2.2.2. Oseltamivir Prescription

Twenty studies (20/57, 35.1%) also reported the change in antiviral prescriptions (oseltamivir). A significant increase in oseltamivir prescription with POCT emerged from 60.0% of the studies (12/20), including one study focused on the implementation of the rapid test for RSV combined with influenza.

Separate meta-analyses stratified by FA-RP and RIDT are reported in Figure 6 and Figure 7. Subgroup analyses are reported in Appendix A.

No significant difference in oseltamivir prescription was noted when comparing FA-RP to the standard test (OR = 0.70, 95% CI 0.22–2.24, *p* = 0.55; I^2^ = 84%) (Figure 6A) [32,33,46,48,51], which is confirmed in the sub-group analysis stratified by country (Europe and the US, see Appendix A). When comparing the use of FA-RP vs. clinical diagnosis of influenza [38,47,49], there was an increased trend in prescriptions but without a significant difference (OR = 2.94, 95% CI 0.48–17.86, *p* = 0.24; I^2^ = 95%) (Figure 6B).

With respect to RIDT, all comparisons from the meta-analysis highlighted a significant increase in oseltamivir prescriptions: the use of RIDT vs. clinical diagnosis [9,21,25,26,28] tripled the odds of having prescribed oseltamivir (OR = 3.25, 95% CI 1.47–7.16, *p* = 0.003; I^2^ = 97%) (Figure 7A), while a more conclusive association emerged when comparing a positive RIDT result vs. negative (OR = 164.21, 95% CI 23.20–1162.49, *p* < 0.001; I^2^ = 57%) (Figure 7B) [12,28,29] and when evaluating a positive RIDT vs. a positive standard test (OR = 9.43, 95% CI 1.65–53.93, *p* = 0.01; I^2^ = 64%) (Figure 7C) [8,12,19].

#### 2.2.3. Length of Stay

Thirty-four studies (34/57, 59.6%) evaluated the length of stay in relation to rapid tests or POCTs, with most of the studies conducted in the hospital setting (20/34 and 7/34 in ED other hospital wards, respectively). Almost half of the studies (16/34, 47.1%) reported a significant reduction in the length of stay with POCTs.

Data were not available from RIDT and FA-RP studies to conduct a meta-analysis.

#### 2.2.4. Days of Therapy

Eighteen different studies (18/57, 31.6%), mostly conducted in the hospital wards (15/18, 83.3%), evaluated a potential change in days of therapy (DOT) related to rapid tests or POCT implementation.

In 61.1% of the studies a reduction in DOT was noted (11/18, 61%).

A meta-analysis was possible only for three observational studies [31,33,41], which compared the use of FA-RP vs. standard tests for the duration of therapy in inpatients. No significant difference was found (see Appendix A). Data from RIDT studies were not available to conduct a meta-analysis for this outcome.

#### 2.2.5. Healthcare Cost

The reduction of healthcare costs was less frequent analyzed; only eight studies, mostly conducted in the ED and hospital wards (6/8, 75.0%), reported data about costs (8/57, 14.0%). In only three studies (3/8, 37.5%) a significant reduction in costs was noted. No meta-analysis was conducted for this outcome.

### 2.3. Quality Assessment of Studies

The NIH Quality Assessment Tool was applied to all 57 studies. As reported in Table 1, 47 studies (82.5%) were assessed as fair, two as poor, and eight as good using a specific tool based on the type of study. Referring to RCT, 75.0% of the studies were considered good (6/8). The remaining two studies were assessed as fair. Considering the observational studies, meant as cohort studies, prospective and retrospective studies, most of the papers were considered as fair (27/29, 93.1%) and one as poor (1/29, 3.4%). Only one study was assessed as good (1/29, 3.4%). Most of the before and after studies were considered fair (18/19, 94.7%), while the other study was assessed as poor (1/19, 5.3%). Figure 8 shows the distribution of the quality assessment for the different types of studies. The complete quality assessment of each study is reported in Appendix A.

## 3. Discussion

Rapid tests and POCTs are important tools to help clinicians discriminate the risk of infection and differentiate between viral and bacterial pathogens. Many rapid tests and POCTs are available nowadays, ranging from quick lab tests to microbiological arrays to even ultrasound bedside imaging protocols.

Advances in technology and recent innovations have allowed diagnostics to enlarge their scope, increase accuracy, minimize the time required for results and, above all, reduce their costs. This leads to the actual usage and applicability of these tests in routine diagnostic evaluation in adults [65]. When efficiently implemented into the diagnostic process, the most valuable characteristic of rapid tests is the shorter time to results, leading to more rapid clinical decisions than the standard test.

Respiratory, gastrointestinal, blood, and cerebrospinal fluid (CSF) arrays have been recently developed and evaluated for the diagnostic workup of outpatient, inpatient, and ED settings also in children, with no univocal interpretation of their utility. Despite these doubts, the potential of the new technologies and new diagnostics is still considered of great value [66].

Indeed, the chance to discriminate the presence of infection with more rapid tests is essential for developing institutional programs to enhance antimicrobial stewardship. As predictable, almost 80% of studies on rapid test implementation were performed after 2007. A further increase occurred during the last two years of the COVID-19 pandemic, with a quarter of studies conducted from 2019 to 2021 since differentiating between viral and bacterial infections has become more crucial [67].

Although rapid tests and POCTs have already been proven valuable and feasible in the adult population, both in outpatient and hospital settings [68,69], studies in children are still lacking.

To our knowledge, this is the first systematic review evaluating rapid tests and POCTs in pediatric settings worldwide and their impact on antimicrobial prescription, healthcare costs, and patient outcomes.

A wide heterogeneity in the use of rapid tests for febrile children with respiratory tract infections emerged. More than 75% of studies reported the implementation of POCTs in hospital settings. In outpatient and primary care practice, Strep A rapid test and influenza tests were the most commonly used POCTs. This is not surprising considering the need for a nearby laboratory to perform other tests such as FA-RP that require different procedures for the analysis of the sample (immunofluorescence, array, molecular probe, etc.). In contrast, the rapid antigen tests represent a quicker (15–30 min) tool that can be undertaken at the bedside. Nevertheless, their use still seems not as widespread compared to the in-hospital setting, and this could also be related to the costs of the POCT and the lack of diagnostic stewardship programs that involve general practitioners and pediatricians working in primary care practices.

Almost 80% of studies on rapid test or POCT usage have been performed in Europe and North America, with only a few in East Asia and none in Africa. Due to their simplicity and quick response, there is wide potential in the implementation of POCTs in low- and middle-income countries and in remote or resource-poor settings with no laboratory infrastructure. However, the simple availability of rapid or easy tests does not ensure their implementation [70,71]. As reported by Pai et al. [72], there are considerable barriers to the widespread use of POCTs at primary levels of the healthcare system, which could be related to the lack of training of physicians or of economic resources to buy them. Nevertheless, it is important to highlight that we did not consider for our review many POCTs that could be more useful in low- and middle-income countries than in high-income countries, such as HIV, HCV, malaria, and tuberculosis POCTs.

Rapid differentiation of respiratory infection in non-critically ill children is essential to positively affect the rate of antibiotic prescriptions. This outcome was assessed for the out-patient setting in 86% of all studies worldwide (80%, 89.5%, and 87.5% of studies in North America, Europe, and Asia, respectively). Regardless of setting, country, and type of test, prescriptions rate seems to be positively affected in 65.3% of cases by rapid tests or POCTs implementation.

More evidence on rapid tests and POCTs for respiratory pathogens in improving both antibacterial and antiviral prescriptions results from the meta-analysis of RIDT and FA-RP.

Different results are reported comparing a positive RIDT result with a negative one or comparing the use of RIDT, regardless of the results, to the clinical diagnosis. In the first case, assessing influenza virus as causative of the clinical presentation of the patients, leads to a reduction in antibiotic prescription. Instead, in the second case, the reduction in antibiotic prescription seems to be less significant. This could be related to different reasons, probably depending on physicians’ prescribing attitudes and intrinsic study limitations.

Even if no definite statement can be made, it is interesting to notice that the trend towards fewer prescriptions is stronger in Europe than in the USA. IDSA guidelines for influenza strengthen the efficacy of an empirical antibacterial therapy to be started in case of deterioration or failure to improve in patients with this diagnosis, especially those with comorbidities [73,74]. Moreover, in the US, antimicrobial stewardship programs have been widespread for longer, so the impact of rapid tests and POCTs in further reducing antibiotics may appear less significant.

Similar results can be extracted from studies evaluating FA-RP, confirming an overall reduction in antibiotic prescriptions. However, as displayed for RIDT, when comparing FA-RP to clinical diagnosis, the evidence for this outcome becomes less strong, despite results from two RCTs assessed as good through the NIH quality assessment rate. Unlike RIDT, this trend appears more significant in the US than in Europe, maybe because the prescription of antibiotics in Europe remains higher for respiratory non-influenza viruses for fear of possible bacterial co-infections [75,76,77]. For example, British guidelines for community-acquired pneumonia (CAP) suggest that all children with a clear clinical diagnosis of CAP should receive antibiotics as bacterial and viral pathogens cannot reliably be distinguished from each other, especially with persistent symptoms [78].

Interestingly, the two studies on the implementation of the rapid test for RSV showed no significant difference in the antibiotic prescription between those testing positive and those negative, suggesting that factors others than the specific etiology determined the decision about prescription. In the study of Thibeault [52], factors associated independently with the cessation of intravenous antibiotics in the case of a positive test for RSV were the age > 3 months and the absence of pneumonia. This is probable due to the fact that young infants are more likely to have severe outcomes than older ones and that the risk of bacterial coinfection could be perceived greater by physicians in case of presence of pneumonia. This is in line with what was found by a Spanish study based on a mail questionnaire, showing that the fear of complications from infections is one of the factors related to the prescribing of antibiotics by general practitioners even if not necessary. Indeed, many physicians agreed that when in doubt as to whether a patient had a bacterial infection, it was better to prescribe a broad-spectrum antibiotic [79,80].

It is important to highlight this point, because it is known that the implementation of these tests alone is often not enough to change antimicrobial prescription patterns [81].

Unfortunately, only six studies included in this review specified whether an antimicrobial stewardship program (ASP) was implemented at the same time of the introduction of the point of care test and in only one the introduction of a rapid test was concomitant with the implementation of an ASP. The implementation of educational interventions combined with the introduction of rapid tests could further improve the antibiotic prescription, as it is also emphasized by a review published in 2018 to ensure rational use of antibiotics [82].

The meta-analysis highlighted as the strongest evidence the improvement of oseltamivir prescription from overall use of RIDT. According to the IDSA guideline on managing seasonal influenza, oseltamivir should be prescribed to patients at high risk of complications from influenza. Regarding children, it is recommended for children aged < 5 years and especially aged < 2 years, children with hematological disease (including sickle cell disease), metabolic disease (diabetes, obesity), immunosuppression, and children and adolescents through 18 years who are receiving aspirin and who might be at risk of Reye Syndrome [73].

Analysis for FA-RP shows a trend toward a more appropriate prescription of antivirals, despite a significant difference. It could be related to the usefulness of this test for detecting a broad spectrum of pathogens rather than influenza only; therefore, this outcome can be limited by less a pre-test probability for clinical, seasonal, or epidemiological reasons.

Unfortunately, no significant data could be extrapolated from RIDT and FA-RP studies for meta-analysis regarding secondary outcomes considered in this review (DOTs, LOS, and healthcare costs). A trend in reduction of DOTs was only evaluated from three studies of FA-RP without a significant difference.

Almost half of the studies in our review reported an overall significant reduction of LOS using the rapid tests, particularly in ED, due to faster clinical decisions. Nevertheless, it is difficult to assess this outcome in hospitalized children because of other possible confounders (e.g., severity of symptoms, comorbidities, complications). For such patients, a decline in the duration of treatment seems to be more relevant. Interestingly, three-quarters of the studies reporting DOTs and LOS as outcomes were conducted in Europe (4/18, 22.2% and 9/34, 26.5%, respectively) and North America (9/18, 50.0% and 17/34, 50.0%, respectively).

The effects of rapid tests and POCTs on healthcare costs have been assessed in only one-third of studies, almost all conducted in hospital settings, with a positive correlation in terms of cost-saving. Whereas other outcomes (rate of prescriptions, LOS, or DOTs) can be easily measured, healthcare cost evaluation involves a more complex analysis to be performed by the authors. Thus, although rapid tests and POCTs should theoretically be cost-saving tools, less evidence emerged from this review that cannot be transferred to outpatient settings.

Finally, considerations can be made from the quality assessment of studies. Most RCTs have been assessed as good using the standardized NIH tool. Only two studies were considered fair, mainly due to an overall drop-out rate higher than 20% and an included population that had not reached the pre-planned sample size. In none of the studies, participants, providers, or people assessing the outcomes were blinded to the intervention. The majority of the observational and before and after studies have been judged as fair. Most of the studies did not have a pre-planned sample size calculation; therefore, it was not possible to assess the statistical power of the studies. Indeed, no studies reported the blinding of the outcome assessors and not all the potentially confounding variables were always measured and adjusted statistically.

It would be desirable that before and after studies, with preplanned sample size calculation or blinded, randomized, clinical trials would be proposed and conducted both in high and low–middle income countries.

### Limitations

The limitations of this study are primarily related to the vast heterogeneity of studies focused on the use of rapid tests and POCTs in pediatric settings. Beyond the type of rapid tests evaluated, another source of variability was due to different brand names used for testing, which involves non-uniform reliability of tests in terms of sensitivity, specificity, and positive or negative predictive value. The authors also did not consider different pre-test probability related to the type of infection for most cases. Moreover, the majority of studies analyzed for this review were set in high-income countries, thus, results and conclusions cannot be broadened to low-resources countries.

All of this was also exacerbated by different modalities for outcome assessment which complicated the comparison between studies.

## 4. Materials and Methods

### 4.1. Study Design and Search Strategy

This systematic review is based on the Preferred Reporting Items for Systematic Reviews and Meta-analyses (PRISMA) guidelines [83]. Embase, MEDLINE, and Cochrane Library databases were systematically searched, combining terms for “children”, “rapid test”, “Point of care test”, and “Outcome Assessment”, with restrictions on dates from 1 January 2000, to 11 September 2021. The full strategy is provided in Appendix A.

The review protocol was registered at the PROSPERO International Prospective Register of Systematic Reviews: Registration Number CRD 42022345124.

### 4.2. Inclusion Criteria and Outcomes

Studies evaluating the effect of the implementation of rapid tests and POCTs for respiratory tract infections that included patients younger than 21 years, both in outpatient or in-hospital settings, were considered for full-text review. Even if the child age is usually considered until 14 years old, we decided to include children until 21 years old because some centers accept these younger adults in the Pediatric Emergency Department.

POCTs were considered those tests for which the sample was analyzed near the patient itself, for example, the Rapid Influenza Diagnostic Test (RIDT) and Strep A test. Instead, rapid tests were those tests for which the sample was collected and then sent to a laboratory that analyzed it and gave results in few hours, for example, the film-array test or PCR for respiratory virus.

Randomized controlled trials, controlled and non-controlled before and after studies, controlled and non-controlled interrupted time series, and cohort studies conducted in all income level countries were included. The primary outcome considered for the final review was the change in the rate of antimicrobial prescriptions (antibacterial and/or antiviral); secondary outcomes included days of therapy (DOTs), length of stay (LOS), and healthcare costs.

### 4.3. Exclusion Criteria

Review, case series, notes and letters, conference abstracts, and opinion articles were excluded. Papers on both adults and children were also excluded if extraction of pediatric data was not possible. Papers on rapid tests and POCTs for HIV, malaria, tuberculosis, hepatitis B or C virus, parasites, or sexually transmitted infections were also excluded. Studies validating rapid tests’ or POCTs’ diagnostic accuracy or only with epidemiological analysis were excluded. Papers on rapid tests on blood, stool, or cerebrospinal fluid were excluded.

### 4.4. Study Selection

In line with the PRISMA guidelines for systemic reviews, titles and abstracts identified through an electronic database were independently screened by three reviewers (GB, AG, MM), and any references which did not meet the inclusion criteria were excluded. For all remaining references, three reviewers obtained full-text copies and independently examined them in detail to determine whether they met all the inclusion criteria for the review. Discussion with a fourth reviewer (DD) resolved any disagreement regarding study selection.

### 4.5. Data Collection

Data were extracted using a standardized data collection form. The following data from each included paper were extracted, where reported: study characteristics (authors, year of publication, study design, study location, multicenter involvement, country); patient characteristics (age, care setting, inclusion and exclusion criteria); type of rapid test or POCT; main results with accuracy measures; outcomes (e.g., rate of antibiotic and antiviral prescription, days of therapy); patient care (e.g., length of stay); economic outcomes.

The income level of each country was defined according to the World Bank List of economies published in July 2021 [7].

### 4.6. Risk of Bias in Individual Studies

Three reviewers independently rated the quality of the included studies using the National Institutes of Health (NIH) Quality Assessment Tool [84]. Based on the type of study, a different NIH tool was used. The overall assessment was good, fair, or poor. The tools with 14 questions were classified as poor if the score was between 0 and 5, fair if between 6 and 10, and good if between 11 and 14. The tools with 12 questions were classified as poor if the score was between 0 and 4, fair if between 5 and 8, and good if between 9 and 12.

### 4.7. Statistical Analysis

When sufficient outcome data were available, we performed meta-analyses. The effect of the intervention was evaluated using the dichotomized measure of odds ratio (OR) and a mean difference (MD) value when different outcome measurements were present for continuous outcomes, along with 95% confidence intervals (CI). Heterogeneity between study-specific estimates was tested using chi-squared statistics and measured with the I2 index (a measure of the percentage variation across the studies caused by heterogeneity) and the Cochran’s Q test [85,86]. A pooled estimate was obtained by fitting the DerSimonian and Laird random-effects model [87], which takes into account both the sampling variance within the studies and the variation in the underlying effect across studies. Separated analyses were performed depending on the study design, observational or randomized controlled studies. Furthermore, subgroup analyses were planned (1) by age group (e.g., less than 5 years or 18 years) and (2) geographical locations (e.g., USA or European countries). All tests were considered significant statistically, for *p* < 0.05. The analyses were performed using Review Manager Version 5.4 (Cochrane Collaboration, London, UK).

## 5. Conclusions

The findings of this systematic review seem to support the implementation of rapid tests and POCTs as a valuable tool to improve antimicrobial prescribing, both reducing unnecessary antibiotics administration and the duration of therapy and increasing appropriate oseltamivir usage. Furthermore, even if it was not possible to perform a metanalysis, the use of rapid tests also seems to be useful in reducing turnaround time and length of stay, particularly in hospital settings.

However, more well-designed studies are still needed to reliably assess the implementation of rapid tests and POCTs in pediatric settings, both in high and low-middle income countries, in order to improve patients’ outcomes and reduce unnecessary prescriptions and healthcare costs.

It is also advocated that implementation of rapid tests should be constantly combined within well-structured antimicrobial stewardship programs, as recommended by international societies for infection and antibiotic resistance control, to improve antimicrobial prescription further.

## Figures and Tables

**Figure 1 antibiotics-11-01192-f001:**
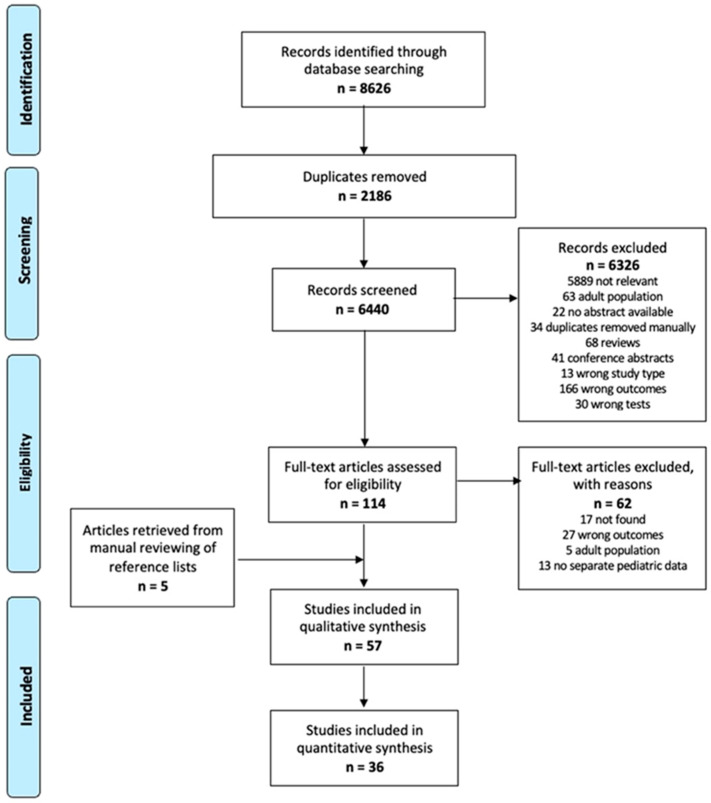
Flowchart of the study selection process (PRISMA).

**Figure 2 antibiotics-11-01192-f002:**
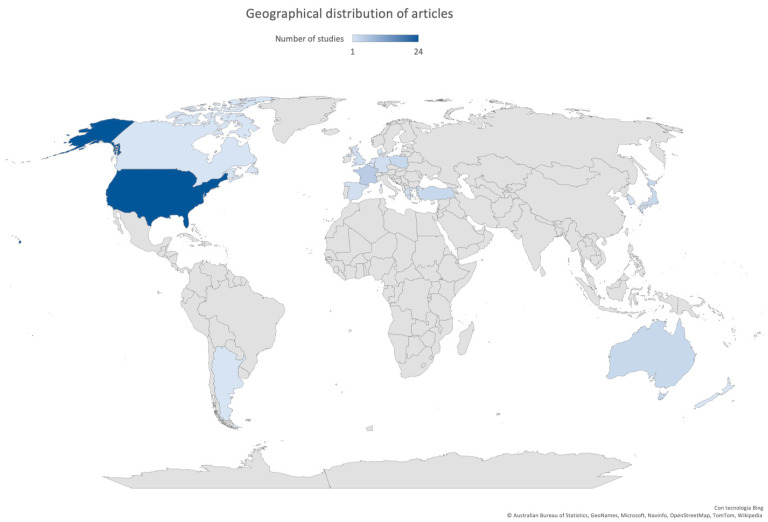
Geographical distribution of articles included in this review.

**Figure 3 antibiotics-11-01192-f003:**
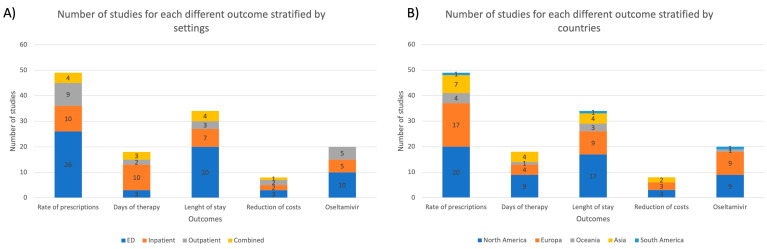
The number of studies for each different outcome stratified by (**A**) settings; (**B**) countries; ED = emergency department.

**Figure 4 antibiotics-11-01192-f004:**
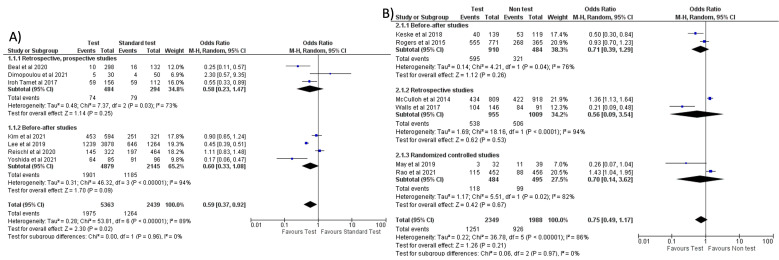
Forest plot of prescription of antibiotics after implementation of FA-RP: (**A**) FA-RP vs. standard test; (**B**) FA-RP versus clinical diagnosis.

**Figure 5 antibiotics-11-01192-f005:**
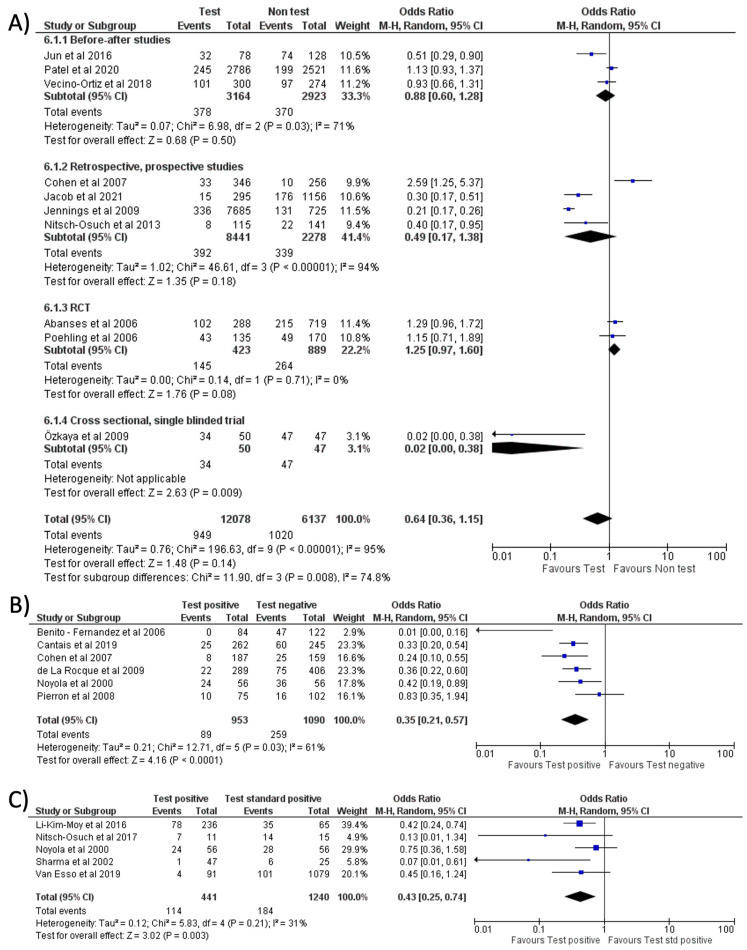
Forest plot of prescription of antibiotics after implementation of RIDT: (**A**) RIDT vs. clinical diagnosis; (**B**) RIDT positive versus RIDT negative; (**C**) RIDT positive vs. standard of care positive.

**Figure 6 antibiotics-11-01192-f006:**
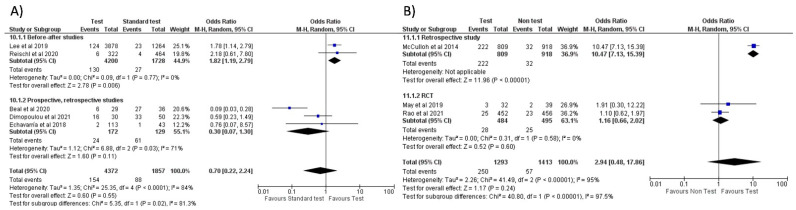
Forest plot of prescription of oseltamivir after implementation of FA-RP: (**A**) FA-RP vs. standard of care; (**B**) FA-RP vs. clinical diagnosis.

**Figure 7 antibiotics-11-01192-f007:**
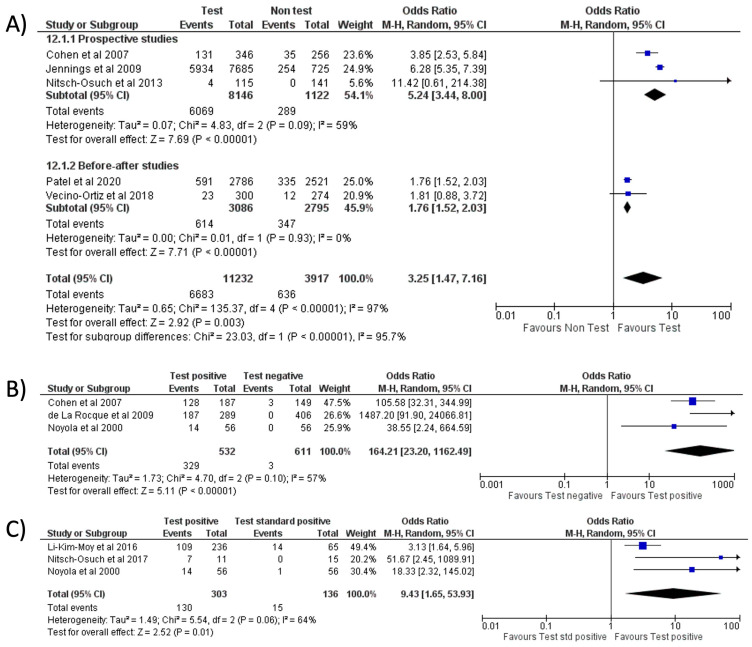
Forest plot of prescription of oseltamivir after implementation of RIDT: (**A**) RIDT vs. clinical diagnosis; (**B**) RIDT positive vs. RIDT negative; (**C**) RIDT positive vs. standard of care positive.

**Figure 8 antibiotics-11-01192-f008:**
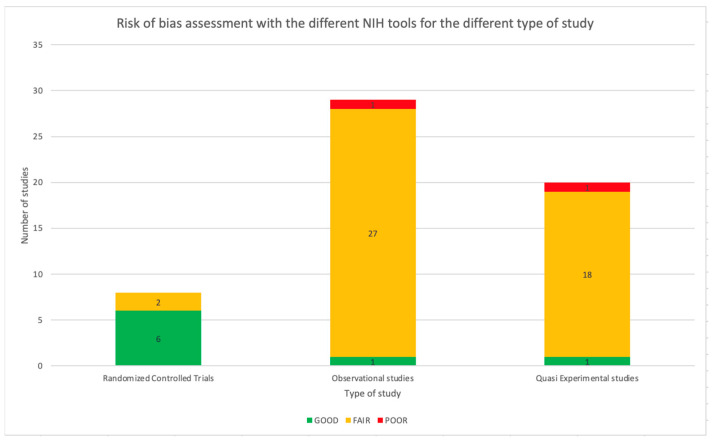
Risk of bias assessment with the different NIH tools for the different types of studies.

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
