# Peer review of "Point-of-Care and Rapid Tests for the Etiological Diagnosis of Respiratory Tract Infections in Children: A Systematic Review and Meta-Analysis"

_antibiotics, 2022, doi:10.3390/antibiotics11091192_

Round 1

Reviewer 1 Report

n.a.

Author Response

Reviewer 1 had no comments.

Reviewer 2 Report

The authors have presented a meta-analysis and systemic review performed according to guidelines on a hugely important pediatric infectious disease topic. Overall the analysis is great and the manuscript is put together fairly well, though several comments are listed below. Particular areas to address are Table 1 and the discussion. I also think the results could be presented more clearly with the addition of a table. There are many grammar issues, which hopefully are easy to correct.

One important item not addressed by this study is the role of diagnostic and antimicrobial stewardship. Many programs have guidelines for how these tests can be used and in some cases stewardship teams manage or provide guidance once tests result. It would be helpful to mention if description of this was provided, and whether or not this was discussed would be something worth mentioning in the discussion. 

Comments by line/section:

Line 64: Many institutions do routinely use rapid tests for evaluating febrile children. Perhaps you mean it isn't standard of care or is less well-studied.

Line 468: However you define these trials, you should be consistent throughout your paper. This would also apply to figure 8, lines 99-101, and table 1. Could simplify before and after and ITS studies to quasi-experimental studies. Also it looks like you have a case-control study, so you may say you included observational instead of cohort. 

Line 499: You mention that this study included high and low-middle income countries only, but this isn't explicitly stated in the methods. You should also provide a brief rationale for that decision. 

Line 505: If you are going through the trouble of describing the point breakdown, it might be worth specifying which tools fell into which group. Alternatively, you can remove these details and list the study tools themselves as a supplement.

Line 512: How did you determine that sufficient outcomes data were available?

Line 523: The subgroup analysis description is confusing based on what you have in the results. First, I don't see any mention of age group in the results. The cutoff of age less than 5 also seems arbitrary and warrants a rationale, but only if you plan to report the results. Additionally, type of country is an odd description. In the results you talk about region but include US separately, and at other times US is referred to in context of North America. It may be best to describe the subgroup as geographical location and list all of those included. 

Table 1 is long and difficult to read. Some specific comments:

- The title column is not necessary since you have the author, citation, and year. You have two different OPs, which you should correct.

- The study design descriptions again need to be clarified and made consistent throughout. I think simplifying them would help a lot.

- Study period probably does not need to be as granular as including the day.

- Zeros and ones don't belong in a table like this; also it's not clear if there is a difference between zero and blank. Better to use Yes/No. Or change to one column with a list of the outcomes for each study. 

- In the methods, rate of prescriptions included antivirals, and oseltamivir prescription rate is not listed as a secondary outcome. 

Lines 156-166: I don't see these data in your figures, thus you may need to summarize in the text. Reporting findings as statistically significant or not can be misleading, as is describing a slight reduction (line 163). 

Line 187/Figure 5A: There are some outliers you may want to mention, in particular, Ozkaya. 

Results: There are a lot of data here, which is summarized well and has good figures but still somewhat burdensome to get through. Some suggestions:

- It would really be helpful to have a table summarizing some of these results.

- A percentage is not needed every time the number of studies is mentioned.

- It may not be necessary to include the citations in the text for every section since you already have them in the figures.

Discussion: This section is not well-written in terms of organization, succinctness, and language/grammar. I think the most important points of your results are identified, but they are a little difficult to pick out and should be emphasized more, particularly for the RIDT results. There are also a few explanations offered regarding your findings that might be strengthened if you could find more literature supporting them (e.g., line 360, line 368, line 379). 

General minor comments on language: I do not believe the test names need to be capitalized unless you are referring to a proprietary name like FilmArray. The words "rapid" and "test" are capitalized throughout. Also, if possible you should have someone else, preferably a native English speaker, read through and correct minor grammar issues, particularly in the discussion.

Author Response

The authors have presented a meta-analysis and systemic review performed according to guidelines on a hugely important pediatric infectious disease topic. Overall the analysis is great and the manuscript is put together fairly well, though several comments are listed below. Particular areas to address are Table 1 and the discussion. I also think the results could be presented more clearly with the addition of a table. There are many grammar issues, which hopefully are easy to correct.

One important item not addressed by this study is the role of diagnostic and antimicrobial stewardship. Many programs have guidelines for how these tests can be used and in some cases stewardship teams manage or provide guidance once tests result. It would be helpful to mention if description of this was provided, and whether or not this was discussed would be something worth mentioning in the discussion.

Thank you for pointing it out; we agree it is an important topic. Unfortunately, only six studies reported if an ASP was implemented at the same time or not, while all the other studies don’t mention it. However, we decided to add a paragraph about this in the discussion.

Comments by line/section:

Line 64: Many institutions do routinely use rapid tests for evaluating febrile children. Perhaps you mean it isn't standard of care or is less well-studied.

Thank you for your advice, we changed the sentence as suggested: “However, their use is not currently the standard of care for evaluating febrile children.”

Line 468: However you define these trials, you should be consistent throughout your paper. This would also apply to figure 8, lines 99-101, and table 1. Could simplify before and after and ITS studies to quasi-experimental studies. Also it looks like you have a case-control study, so you may say you included observational instead of cohort.

Thank you for your advice; we changed the study types into three groups: “randomized control trials”, “observational studies”, and “quasi-experimental studies”. We also changed Figure 8: even if the tools used to evaluate before and after studies and case-control studies were different, they comprehended 12 questions, so they were similar in calculating the result.

Line 499: You mention that this study included high and low-middle income countries only, but this isn't explicitly stated in the methods. You should also provide a brief rationale for that decision.

We included all the studies we found with our research, regardless of the economic status defined according to the World Bank List of economies published in July 2021, as explained in the Material and Methods section. To clarify, we added that we included the studies conducted in all income level countries.

Line 505: If you are going through the trouble of describing the point breakdown, it might be worth specifying which tools fell into which group. Alternatively, you can remove these details and list the study tools themselves as a supplement.

Thank you for your advice. We clarified in the section which tools fell into which group: “The tools with 14 questions (Observational studies and Randomized Control trial) were classified as poor if the score was between 0 and 5, fair if between 6 and 10, and good if between 11 and 14. The tools with 12 questions (Before and after and Case Control Studies) were classified as poor if the score was between 0 and 4, fair if between 5 and 8, and good if between 9 and 12.”

Line 512: How did you determine that sufficient outcomes data were available?

We considered the presence of at least three articles for each outcome sufficient to conduct a metanalysis.

Line 523: The subgroup analysis description is confusing based on what you have in the results. First, I don't see any mention of age group in the results. The cutoff of age less than 5 also seems arbitrary and warrants a rationale, but only if you plan to report the results. Additionally, type of country is an odd description. In the results you talk about region but include US separately, and at other times US is referred to in context of North America. It may be best to describe the subgroup as geographical location and list all of those included.

Thank you for pointing it out. We would like to conduct a subgroup metanalysis by age group to analyze and eventually report the difference in antibiotic prescription patterns in preschool children compared to older children (which is why we choose five years as the cut-off). However, we couldn’t conduct this pre-planned analysis due to a lack of data. To clarify it, we add a sentence in the result section.

Regarding the type of country, we changed the “type of country” to “geographical location” to avoid misunderstanding.

Table 1 is long and difficult to read. Some specific comments:

- The title column is not necessary since you have the author, citation, and year. You have two different OPs, which you should correct.

We deleted the title column as suggested. We clarified the two different Ops. OP means Observational Prospective while PO means Prescription of Oseltamivir. Sorry for the misprint

- The study design descriptions again need to be clarified and made consistent throughout. I think simplifying them would help a lot.

We clarified the study description and made it consistent throughout as mentioned above

- Study period probably does not need to be as granular as including the day.

We changed the study period including only months and years

- Zeros and ones don't belong in a table like this; also it's not clear if there is a difference between zero and blank. Better to use Yes/No. Or change to one column with a list of the outcomes for each study.

Thank you for your advice. We decided to divide the table: Table 1 represents the characteristic of the studies included in the review, with the outcome considered by each study (labelled with an X in the specific column); Table 2 represents the specific outcome, with the results reported in each outcome.

- In the methods, rate of prescriptions included antivirals, and oseltamivir prescription rate is not listed as a secondary outcome.

We added oseltamivir prescription as a secondary outcome

Lines 156-166: I don't see these data in your figures, thus you may need to summarize in the text. Reporting findings as statistically significant or not can be misleading, as is describing a slight reduction (line 163).

To better clarify this section, we added the results of each study in Table 2.

Line 187/Figure 5A: There are some outliers you may want to mention, in particular, Ozkaya.

Thank you for your advice; we mention the study suggested in our results.

Results: There are a lot of data here, which is summarized well and has good figures but still somewhat burdensome to get through. Some suggestions:

- It would really be helpful to have a table summarizing some of these results.

We included a second Table with all the results

- A percentage is not needed every time the number of studies is mentioned.

Thank you for your advice, we made this improvement.

- It may not be necessary to include the citations in the text for every section since you already have them in the figures.

Thank you for your advice; we made this improvement. We preferred to specify the citation in the section about meta-analysis because there were no citations in the figure.

Discussion: This section is not well-written in terms of organization, succinctness, and language/grammar. I think the most important points of your results are identified, but they are a little difficult to pick out and should be emphasized more, particularly for the RIDT results. There are also a few explanations offered regarding your findings that might be strengthened if you could find more literature supporting them (e.g., line 360, line 368, line 379).

Thank you for pointing it out. We tried to modify the discussion section and emphasize the results. We searched for more literature to support our findings.

General minor comments on language: I do not believe the test names need to be capitalized unless you are referring to a proprietary name like FilmArray. The words "rapid" and "test" are capitalized throughout. Also, if possible you should have someone else, preferably a native English speaker, read through and correct minor grammar issues, particularly in the discussion.

We changed all the test names throughout the text.

Reviewer 3 Report

1.The manuscript title of diagnostic stewardship and the final conclusions/recommendations do not seem to be well aligned. Suggest removal of the term diagnostic stewardship.

2. To include in the title the term Europe-USA or high income countries since the manuscript conclusions which are derived from publications analyzed seem to pertain to Europe-USA and the findings/conclusions are relevant to these countries only.

3.Under study inclusion criteria-The authors need to justify why subjects upto age 21 were included in the analysis since these are clearly outside the pediatric age group.

3.The findings of the study as applied to antibiotic prescriptions seem to broad. It would be helpful if the study can pinpoint to specific antibiotic such as augmentin which is widely used to treat respiratory infections in children.

4.Discussion line 348- 355 is not very clear. What are the authors trying to convey?

5.The discussion is very long with repetition of facts, can be condensed.

6.Under study limitations it should be clearly stated that the study conclusions pertains only to high income countries and cannot be generilized to other settings.

7.Conclusion-Line 539 spelling of rapid tests- please correct.

Author Response

1.The manuscript title of diagnostic stewardship and the final conclusions/recommendations do not seem to be well aligned. Suggest removal of the term diagnostic stewardship.

Thank you for your advice; we changed the title as suggested: “Point-of-care and Rapid tests for the etiological diagnosis of respiratory tract infections in children: a Systematic Review and Meta-analysis.”

2. To include in the title the term Europe-USA or high income countries since the manuscript conclusions which are derived from publications analyzed seem to pertain to Europe-USA and the findings/conclusions are relevant to these countries only.

Thank you for your advice. Even if the manuscript conclusions are not generalizable to low and middle-income countries, we decided not to specify them in the title because we also included some articles from low and middle-income countries.

3. Under study inclusion criteria-The authors need to justify why subjects upto age 21 were included in the analysis since these are clearly outside the pediatric age group.

We explained it in the text: “Even if the child age is usually considered until 14 years old, we decided to include children until 21 years old because some centers accept these younger adults in Pediatric Emergency Department.”

4. The findings of the study as applied to antibiotic prescriptions seem to broad. It would be helpful if the study can pinpoint to specific antibiotic such as augmentin which is widely used to treat respiratory infections in children.

Thank you for your advice, we agree with you. Unfortunately, only a few studies specified the type of antibiotics administered, while the majority of the studies considered only the antibiotic prescription, without specifying if narrow or broad antibiotics and the specific type of antibiotic

5. Discussion line 348- 355 is not very clear. What are the authors trying to convey?

We tried to clarify it better in the text: More evidence on Rapid Tests and POCT for respiratory pathogens in improving antimicrobial prescriptions (both antibacterial and antiviral) results from the meta-analysis of RIDT and FA-RP.

Different results are reported comparing a positive RIDT result with a negative one or comparing the use of RIDT, regardless of the results, to the clinical diagnosis. In the first case, assessing influenza virus as causative of the clinical presentation of the patients led to a reduction in antibiotic prescriptions. Instead, in the second case, the reduction in antibiotic prescription seems to be less significant. This could be related to different reasons, probably depending on physicians’ prescribing attitudes and intrinsic study limitations

6. The discussion is very long with repetition of facts, can be condensed.

We thank the reviewer for this advice. Please find a revised and condensed version of the Discussion paragraph in the full-text.

7. Under study limitations it should be clearly stated that the study conclusions pertains only to high income countries and cannot be generilized to other settings.

Thank you for pointing it out. We implemented our statement as follows:

“The limitations of this study are primarily related to the vast heterogeneity of studies focused on the use of Rapid tests and POCTs in pediatric settings. Beyond the type of Rapid Tests evaluated, another source of variability was due to different brand names used for testing, which involves non-uniform reliability in terms of sensitivity, specificity and positive or negative predictive value. The authors also did not consider different pre-test probabilities related to the type of infection for most cases. Moreover, the majority of studies analyzed for this review are set in high-income countries; thus, results and conclusions can’t be broadened to low-resources countries”.                          

8. Conclusion-Line 539 spelling of rapid tests- please correct.

Thank you and sorry for the misprint, we corrected it within the text (“Rapid Tests”).